# Compressive Behaviour of Closed-Cell Aluminium Foam at Different Strain Rates

**DOI:** 10.3390/ma12244108

**Published:** 2019-12-09

**Authors:** Nejc Novak, Matej Vesenjak, Isabel Duarte, Shigeru Tanaka, Kazuyuki Hokamoto, Lovre Krstulović-Opara, Baoqiao Guo, Pengwan Chen, Zoran Ren

**Affiliations:** 1Faculty of Mechanical Engineering, University of Maribor, 2000 Maribor, Slovenia; 2Department of Mechanical Engineering, TEMA, University of Aveiro, 3810-193 Aveiro, Portugal; 3Institute of Pulsed Power Science, Kumamoto University, Kumamoto 860-8555, Japan; 4Mechanical Engineering and Naval Architecture, Faculty of Electrical Engineering, University of Split, 21000 Split, Croatia; 5State Key Laboratory of Explosion Science and Technology, Beijing Institute of Technology, Beijing 100811, China

**Keywords:** cellular materials, closed-cell aluminium foam, quasi-static, high strain rate, powder gun, computational simulations, crushable foam

## Abstract

Closed-cell aluminium foams were fabricated and characterised at different strain rates. Quasi-static and high strain rate experimental compression testing was performed using a universal servo-hydraulic testing machine and powder gun. The experimental results show a large influence of strain rate hardening on mechanical properties, which contributes to significant quasi-linear enhancement of energy absorption capabilities at high strain rates. The results of experimental testing were further used for the determination of critical deformation velocities and validation of the proposed computational model. A simple computational model with homogenised crushable foam material model shows good correlation between the experimental and computational results at analysed strain rates. The computational model offers efficient (simple, fast and accurate) analysis of high strain rate deformation behaviour of a closed-cell aluminium foam at different loading velocities.

## 1. Introduction

Metal foams and cellular structures are being widely studied in different aspects and considered for use in modern applications in engineering, medicine, fashion and others recently [1,2]. They are one of the most promising building materials for future products due to their attractive combination of low density, high energy absorption capability, damping and thermal insulation [3].

Different fabrication procedures were developed in the last decades and most of them are briefly presented in [4]. Their significant cost is still the most important limitation for the mass production of foam and cellular structures. The additive manufacturing technologies are an increasingly powerful tool for fabrication of different types of cellular structures with exactly predefined and controlled geometry on different scales in recent years [5,6,7]. Some cheaper ways to produce cellular structures with additive manufacturing techniques have been developed recently [8]. Alternatively, a powder metallurgy is already a well-established procedure for the fabrication of closed-cell foams [9,10]. The foams fabricated this way have a characteristic dense material surface layer completely surrounding the inner closed-foam structure. Despite such foams showing a ductile behaviour appropriate for structural applications, this process does not allow for a rigorous control of the cell structure in terms of pore shape and size [4]. This is mainly due to the difficulty of coordination between the mechanisms of metal melting and thermal decomposition of the blowing agent [4]. The problems associated with the quality of these foams and the manufacturing reproducibility has been studied and discussed [11,12,13].

The metallic foams can be divided into two major groups when considering the morphology of the cells: the open- and the closed-cell foams [7]. The open-cell foams are used for functional applications (e.g., filtration, catalysis, heat transfer and biomedical applications), while closed-cell foams are mostly being used for structural applications, typically as weight-saving and impact-absorbing structures in vehicles.

Since this work is concerned only with the closed-cell foams, the following review of work done previously only on this topic is given briefly in the following. The mechanical behaviour of closed-cell aluminium foams was thoroughly characterised under unconstrained and constrained compression loading conditions [14]. The same foams were also inserted in tubes (ex-situ foam filled tubes) and tested under compression and bending loading conditions [15,16]. The powder metallurgy foam fabrication enables also fabrication of closed-cell foams inside the tubes (in-situ foam filled tubes) to achieve increased stiffness by better bonding between the foam filler and the outer tube [17]. The behaviour of closed-cell foams under different loading conditions was also studied in some detail [18], with micro computed tomography (μCT) applied to track the changes of internal structure during the deformation of closed-cell aluminium foam [19].

The influence of the loading velocity up to 150 m/s on the mechanical response of additively manufactured lattice materials [20] and closed-cell aluminium foams [21] was analysed with the Split Hopkinson Pressure Bar (SHPB) apparatus. High strain rate compressive behaviour of aluminium alloy foams was also studied and discussed in [22,23]. Authors in [24] studied the effect of porosity and differences in the deformation pattern of closed-cell aluminium foam under quasi-static and dynamic (SHPB) loading. All mentioned studies point out that the failure modes under quasi-static and dynamic (high strain rate) loading are different. This is attributed to the inertial effect and material strain-rate sensitivity [22,25,26].

The change of the deformation mode and high strain rate hardening was also observed in computational simulations of metallic foams. Mesoscopic computational simulations were performed using μCT scan based geometrical models [24,27]. The Arbitrary Lagrange Eulerian (ALE) method was also applied to take into account the entrapped air inside the closed-cell foam [28]. A cell-based approach also confirmed deformation localisation at higher strain rates [29]. Computational studies of closed-cell aluminium foam and cell-based Voronoi lattice dynamic behaviour showed relationships between the cell geometry and loading rate [30,31]. The behaviour of closed-cell foams can be successfully described by homogenised computational models, i.e., crushable foam material models, which were successfully applied for polymeric, auxetic and aluminium foam core of composite panels under different loading conditions [32,33,34]. Besides closed-cell foams, similar work was done in the field of open-cell foams in terms of multiaxial loading [35] and impact response [36,37].

All mentioned studies observe that the high strain rate hardening appears above certain (limiting) loading velocity, which is mostly a consequence of inertia effects. An extensive review of dynamic compressive behaviour of cellular materials is given in [38]. As proven many times before, the deformation mode of a given cellular material or structure changes with increasing loading velocity (strain rate). The deformation behaviour of cellular materials at different loading velocities can be in general divided into three deformation modes: homogeneous, transition and shock deformation mode [23,39,40,41]. These three deformation modes are separated by two critical loading velocities.

This work focuses on experimental and computational evaluation of high strain rate behaviour of closed-cell aluminium foam. The results of experiments and validated computational models present the basis for high strain rate deformation and hardening analysis in combination with the analytical approach. The deformation modes and critical velocities of the closed-cell aluminium foam were determined, evaluated and are discussed herein. There is limited experimental and computational work done in the field of analysis of the deformation modes of closed-cell aluminium foams. Therefore, this work provides insight into the deformation behaviour of interesting lightweight materials which will be used in modern constructions.

## 2. Fabrication of Specimens and Experimental Methods

The base closed-cell aluminium foam was fabricated using the powder metallurgy method [10], which consisted of heating extruded foamable precursor material placed into a cylindrical stainless steel mould in a pre-heated furnace (MJAmaral, Vale Cambra, Portugal, Figure 1a) at 700 °C for 12 min. The cavity of the cylindrical mould with inner diameter of ~25 mm and length of ~150 mm was fully filled by the formed liquid metallic foam in the process by foaming the precursor material (Figure 1b) with mass of 61 g and made of aluminium, silicon (7 wt.%) and titanium hydride (0.5 wt.%). The precursor material was prepared by cold isostatic pressing (Schunk Sintermetalltechnik GmbH, Gießen, Germany) of the powder mixture, followed by extrusion through a horizontal 25 MN direct extrusion machine (Honsel AG, Meschede, Germany), resulting in a rectangular bar of 20 mm × 5 mm in cross-section [9]. The mould with formed foam was extracted from the furnace and cooled down to room temperature after heating time. The cylindrical aluminium foam (Figure 1c) was then removed from the mould made of S235JR carbon steel (0.17% C, 1.40% Mn, 0.045% P, 0.045% S, 0.009% N) and cut longitudinally to six equal pieces to prepare the cylindrical specimens (Figure 1d) for mechanical testing. The average pore diameter of the resulting closed-cell foams is approximately 2.65 mm (standard deviation: ~0.87 mm). The specimen’s data are given in Table 1.

Uniaxial compression tests were performed under quasi-static loading conditions using a servo-hydraulic INSTRON 8801 testing machine with position controlled cross-head rate of 0.1 mm/s. The tests were carried out according to the standard ISO 13314: 2011 [42]. In order to reduce the friction and thus minimise the support/loading boundary effects on specimens, the loading plates of the testing machine were lubricated. The engineering stress-strain data were then calculated using the initial specimen’s dimensions from the recorded force-displacement data.

The high strain rate behaviour of closed-cell foams was studied by using a one-stage powder gun which is capable of accelerating a projectile up to velocity of 1.5 km/s. The projectile is, in the case of the powder gun, accelerated by the combustion gas of gun powder detonation. This method was already successfully used for the determination of high strain rate behaviour of other cellular materials including auxetic cellular structures [20,43]. The powder gun device is assembled from two chambers (explosion and target) which are connected with a barrel with an inner diameter of 40 mm. Both chambers are decompressed with the vacuum pump to near vacuum, in order to minimise the influence of the air resistance during the tests. In the target chamber, there is an optical observation window, which enables the visual observation of the deformation procedure, and also an electrical terminal, which enables processing of the signal during an impact. The rigid wall and shock are also positioned in the target chamber, where the impact of cellular structure and sabot is happening, and offers rigid support during the impact of cellular structure, absorbing the energy of the impacting projectile afterwards. The projectile sabot was made from polyethylene (UHMWPE) with a brass weight mounted to the front of the sabot as a projectile driver (total weight of the projectile was about 180 g). The closed-cell foam was positioned and glued with epoxy resin to the brass weight (Figure 2), which enables control of the impact velocity. The projectile’s velocity for this testing was set to 270 m/s, which resulted in the engineering strain rate of approximately 12,000 s^−1^.

The impact velocity and the deformation behaviour of the closed-cell aluminium foam during impact were captured by the HPV-1 high-speed video camera (produced by SHIMADZU corporation, image capture number: 100, maximum resolution: 1 μs). The mechanical response of closed-cell foams (impact pressure) of the closed-cell aluminium foam on the rigid wall was measured with the PVDF gauge (Piezo film stress gauge, PVF2 11-, 125EK, Dynasen), which was already successfully used in previous experiments [43].

## 3. Experimental Results

### 3.1. Quasi-Static Testing

The results of quasi-static (Q-S) compression testing in the form of observed engineering stress-strain relationship show a characteristic compressive behaviour of cellular materials, Figure 3. After an initial quasi-linear response, the cell walls start to bend and buckle resulting in the stress plateau, which is typical for cellular materials [44]. The cell walls begin to fracture with further deformation and the cells gradually collapse, finally leading to the densification of the cellular structure. The resulting deviation is a consequence of the aluminium foam density variation throughout the specimens (628–713 kg/m^3^), which strongly influences the mechanical behaviour. Studies have demonstrated that these foams develop imperfections and structural defects (e.g., micropores) during their fabrication, creating weaker regions where the foam starts to deform, developing one or more visible deformation bands perpendicular to the loading direction [25,45]. The stiffness of the specimens increases by increasing the foam density. Also, the plateau region of these specimens is slightly inclined. This agrees with our previous results [9,12] for these types of foams. As can be seen from Figure 4, the deformation behaviour is uniform at lower strains, which is followed by crushing in shear planes that are formed in the areas with less stiff cellular structure.

### 3.2. High Strain Rate Testing

The results of high strain rate (HSR) testing in the form of observed engineering stress-strain relationship are shown in Figure 5. Again, the response is typical for cellular structures with three distinct differences due to high velocity loading: (i) initial stress peak, (ii) stress oscillations during stress plateau region, and (iii) abrupt densification (no smooth transition between the plateau stress region and densification region, as is usual at quasi-static loading).

The deformation pattern captured with a high speed camera is shown on Figure 6, where it can be observed that the deformation mode changed to shock mode in comparison to the quasi-static response. Consequently, most of the deformation happens at the impact front between the rigid wall and specimen until the specimen is completely compressed.

The comparison between the quasi-static and high strain rate responses is shown in Figure 7. A significant strain rate hardening effect can be observed in the case of high strain rate loading, which is mostly the consequence of the changed deformation mode and micro-inertia effects. The change in the deformation mode is further analysed in Section 4.3.

### 3.3. The Specific Energy Absorption Analysis

The specific energy absorption (SEA) was analysed to evaluate the influence of the strain rate hardening. The SEA was calculated by dividing the absorbed energy (integral under average stress-strain curve) by crushed mass and results for both loading cases are given in Table 2. The high strain rate SEA (evaluated at strains of 0.50 and 0.77) increased by 191.7% in comparison to the quasi-static response at the strain of 0.77 (350% at strain of 0.50), which illustrates significant enhancement potential in the energy absorption capabilities of tested foams for use in applications characterised by high strain rate of deformation.

The evolution of SEA and the discrepancy of all experimental results in the case of quasi-static and high strain rate loading are shown in Figure 8a. Quasi-linear increase of SEA in the case of high strain rate loading can be observed, while in the case of quasi-static loading the increase in the SEA at larger deformations becomes progressively exponential. The discrepancy of the experimental results is also shown in Figure 8a, which is smaller at lower strain rates. Figure 8 makes it possible to determine the specific energy absorption capabilities of closed-cell aluminium foam at different strains and strain rates. The comparison in terms of energy absorption between the experimental results and different types of cellular metals given in the literature [3] is shown in Figure 8b. Good agreement in terms of deformation energy can be observed for the density of cellular material used in this research.

## 4. Computational Simulations

### 4.1. Computational Model

The computational model with applied homogenised material model (crushable foam) with volumetric hardening was built in Abaqus Finite Element software, where the explicit solver was used for analysis. The crushable foam model in Abaqus is based on the flow rule, which is non-associated [46]. The tensile hydrostatic stress is kept constant through the deformation process, while the compression hydrostatic stress changes due to the densification and collapse of the cellular structure. Crushable foam material model with volumetric hardening is based on the experimental observation with a significantly different deformation behaviour under the tensile and compression loading conditions. In the case of compression loading, the capability of volumetric deformation is much larger due to the buckling of struts in the cellular structure [47]. The deformation of crushable foam is by default irreversible, so it can be in most cases treated as plastic deformation. This allows us to define the hardening curve of the material model only with the values of uniaxial compression loading in respect to uniaxial plastic deformation. The hardening curve was determined using a genetic optimisation algorithm (OptiMax software developed at University of Maribor) during the validation procedure [48], with the definition of the material parameters provided in Table 3 and Table 4.

Axial symmetry with axisymmetric boundary conditions could be applied in the computational model due to the axial symmetry of the specimens (Figure 9). The loading (top) and support (bottom) plates were modelled with interaction, where all boundary nodes on the top and bottom edge were connected to two reference points using kinematic coupling, respectively. Boundary conditions were then prescribed to these two reference points. The loading plate was prescribed with a constant velocity of 1 m/s (quasi-static testing) and 270 m/s (high strain rate testing). The higher computational loading velocity in respect to the experimental quasi-static velocity was determined by comparing the reaction forces at the loading and support plates, which should be identical if there are no inertia effects present. The support plate has constrained displacement in *y*-direction and constrained rotation around *z*-axis, while the loading plate has constrained only the rotation around *z*-axis.

The finite element mesh consists of 299 linear quadrilateral elements (type CAX4R), with average global size of 1 mm. The appropriate size of the finite elements was determined during the convergence study with three different finite element sizes.

### 4.2. Computational Results and Comparison with the Experimental Observations

The crushable foam constitutive model parameters were determined by using an optimisation algorithm, initially comparing the quasi-static experiment and computational responses (Figure 10). The material parameters are given in Table 3, where *ρ* is density, *E* is Young’s modulus, *ν* is Poisson’s ratio, *k* is compression yield stress ratio and *k*_t_ is hydrostatic yield stress ratio. After optimisation of the material parameters, the same material model was used for the high strain rate loading. Good agreement can be observed for both quasi-static and high strain rate testing. Thus, the computational model was successfully validated.

The agreement between the experimental and computational responses is very good at both analysed strain rates (Figure 10). A discrepancy can be observed only at lower strains of the HSR response, and only the first stress peak could not be observed in the computational model due to initial boundary conditions. The stress peak is a consequence of collision in the experiments and represents a typical response of structures during the initial phase of the impact, but does not affect the global behaviour afterwards [20].

Additionally, the SEA capabilities from experimental results and computational simulations were compared. The SEA values from experimental tests are given in Table 2, while the SEA values from computational simulations are 32.19 J/g in the case of Q-S loading and 95.11 J/g in the case of HSR loading. This results in 10% overestimation of the energy absorption capabilities calculated by computational simulations, which is caused mainly by a discrepancy in the densification region in the case of Q-S loading and due to the oscillations (no filer has been used) at the plateau stress region in the case of HSR loading.

The deformation behaviour in the case of quasi-static and high strain rate loading conditions is shown in Figure 11. In the case of quasi-static loading, a uniform deformation of the specimen can be noted. The change in the deformation mode can be observed in the case of high strain rate loading, where the deformation is localised at the impact front between loading plate and specimen. This is a consequence of inertia effects and is analysed in detail in Section 4.3.

### 4.3. High Strain Rate Behaviour Analysis

The change of the deformation mode is a consequence of the inertia effect and is triggered at so-called critical velocities. The deformation mode is homogeneous at loading velocities below the first critical velocity. A transition deformation mode is observed between the first and the second critical velocity. The deformation mode changes to shock mode above the second critical velocity, and is characterised by concentrated deformation at the impact front [40,41,49]. Herein, the critical loading velocities of the analysed closed-cell aluminium foam were determined from experimental testing and computational simulation results, described in the previous section.

Dynamic deformation behaviour of cellular (porous) materials can be described by several constitutive crushing foam models, which are presented in [38]. The Rigid-Power-Law Hardening (R-PLH) model was chosen in this study since it enables high-precision predictions of shock-induced stress also in the densification region. The first critical loading velocity can then be obtained as [23]:(1)$$v_{cr1}={\left[\sigma_{0}/9K\right]}^{\frac{n+1}{2n}}\sqrt{\frac{K}{\rho_{0}}}$$
where *σ*_0_ is the initial yield stress (calculated as average stress between 0.2 and 0.4 of strain), *K* is the strength index, *n* the strain hardening index and *ε*_d_ the densification strain. The second critical loading velocity is defined as [23]:(2)$$v_{cr2}=\sqrt{\frac{K}{\rho_{0}}}\varepsilon_{D}^{\frac{n+1}{2}}$$

The material parameters of R-PLH model (*σ*_0_, *K*, *n* and *ε*_d_) were fitted according to the experimental data from quasi-static closed-cell aluminium foam compression tests (Figure 12) and are given in Table 5 together with computed critical velocities. The fitting was done in MS Excel software and nonlinear GRG solving method [50].

The parametric computational study of closed-cell aluminium foam behaviour at different loading velocities (30, 40 and 50 m/s) was then performed to evaluate and additionally validate the developed computational model. Figure 13 shows the mechanical response of closed-cell aluminium foam calculated from computed reaction forces on loading (top) and support (bottom) loading plates. Figure 13a shows the recorded response at loading velocity of 30 m/s (below first critical velocity) where responses at top and bottom loading plates are very similar. This indicates the homogeneous deformation mode under quasi-static loading where no significant inertia effects are observed, which is manifested in similar reaction forces on top and bottom plates. In the case of loading velocity of 40 m/s (Figure 13b), the major discrepancy between the responses is already observed at larger strains, while only a minor discrepancy is observed at strains up to 0.6. At a velocity of 50 m/s (higher than the second critical velocity) (Figure 13c), the discrepancy between responses on the bottom and top plate is even more significant. With further increase of the loading velocity, the discrepancy is progressively increasing, which was clearly observed also at achieved velocities during experiments (Figure 11).

The change in the reaction forces is a consequence of inertia and internal deformation of the foam, which is concentrated in the area of the impact in the case of high strain rate loading. This can be clearly seen in Figure 14, where detailed analysis of effective plastic strain evolution at different loading velocities is shown. In the case of loading velocity of 20 m/s (Figure 14a), the effective plastic strain evolution is more or less homogeneous, while in the case of velocities larger than 30 m/s (Figure 14b–d), the effective plastic strain is localised at the impact front.

## 5. Conclusions

Cylindrical closed-cell aluminium foam specimens were fabricated with the powder metallurgy method and subjected to experimental testing under quasi-static and dynamic loading conditions, where significant strain rate hardening was observed. The high strain rate testing was performed at the strain rates above 12,000 s^−1^, which results in shock deformation mode. The analysis of specific energy absorption capabilities has shown that the energy absorption of the aluminium foam is almost 200% larger at higher strain rates (impact) in comparison to quasi-static loading, which should be considered in the design of modern products and structures incorporating closed-cell aluminium foam.

A homogenised crushable foam computational model was developed and validated with experimental results. The developed axisymmetric homogenised computational model offers fast, robust and accurate prediction of the mechanical response of closed-cell aluminium foam. The validation of computational models enabled further relevant study of closed-cell aluminium foam behaviour at different strain rates. First, the critical velocities separating different deformation modes (homogeneous, intermediate and shock mode) were determined by using quasi-static experimental data and the Rigid-Power-Law Hardening (R-PLH) model. It was computationally confirmed that the change of the deformation mode appears at the same critical velocities, which were analytically determined. Furthermore, study of the mechanical responses at the loading and support plates was also performed, which proved that the discrepancy between the responses is getting more significant at higher loading velocities above the first critical velocity, where inertia effects are more pronounced.

Results presented in this study clearly indicate the advantages of using closed-cell aluminium foam in different dynamic and impact applications, e.g., crashworthiness, ballistic and blast protection, due to the strain rate hardening and strain rate sensitivity. Their excellent energy absorption capabilities allow absorption of more energy at larger strain rates due to the change in the deformation mode and localised deformation. Furthermore, the proposed computational model is efficient (simple, fast and accurate) and offers a possibility to study the behaviour of closed-cell foam in larger composite panels or tubes in possible industrial applications.

## Figures and Tables

**Figure 1 materials-12-04108-f001:**
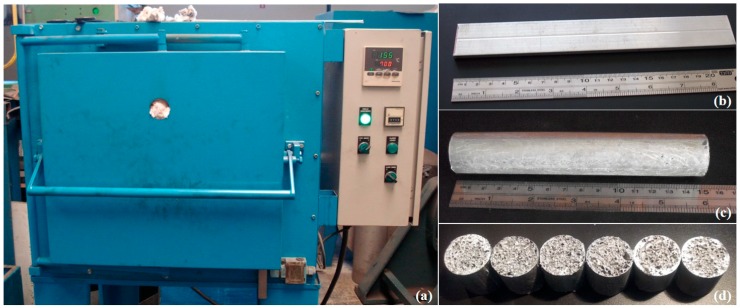
The foaming furnace (**a**) and precursor material (**b**) used in this work, as well as the visual aspect of the resulting cylindrical aluminium foam (**c**) and the closed-cell foam specimens for mechanical testing (**d**).

**Figure 2 materials-12-04108-f002:**
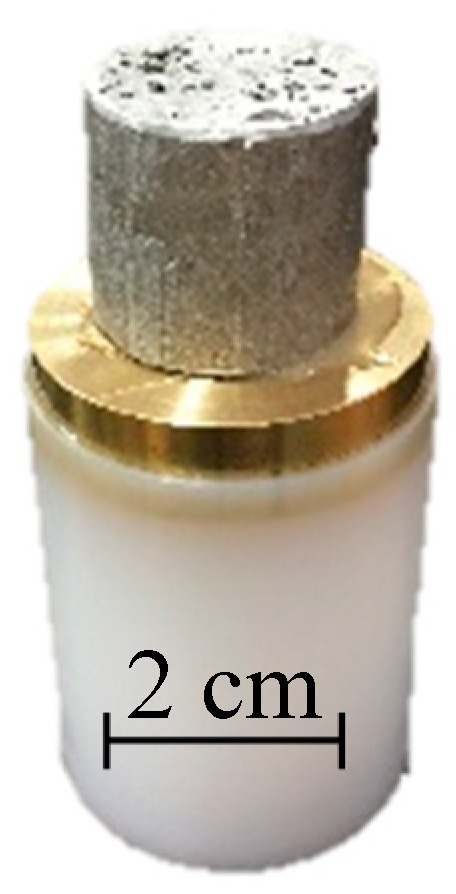
Closed-cell foam specimen mounted on projectile.

**Figure 3 materials-12-04108-f003:**
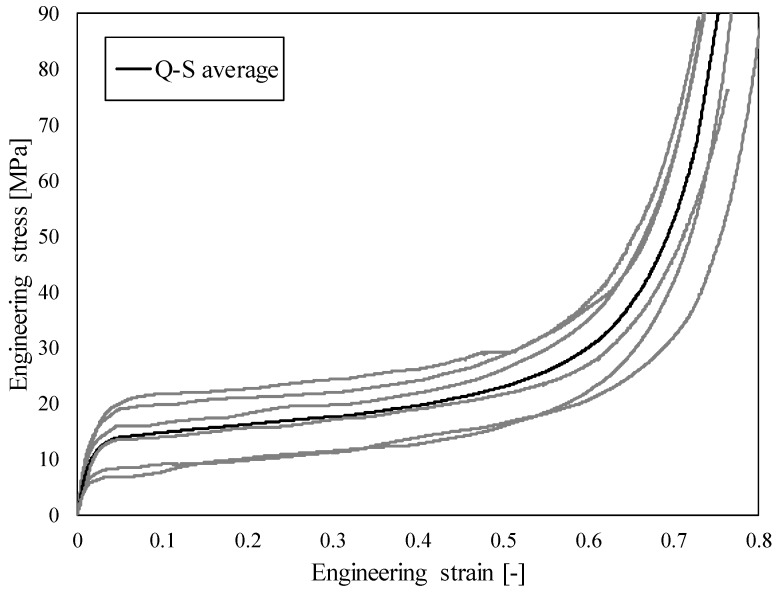
Engineering stress-strain relationship of closed-cell aluminium foam under quasi-static loading conditions.

**Figure 4 materials-12-04108-f004:**
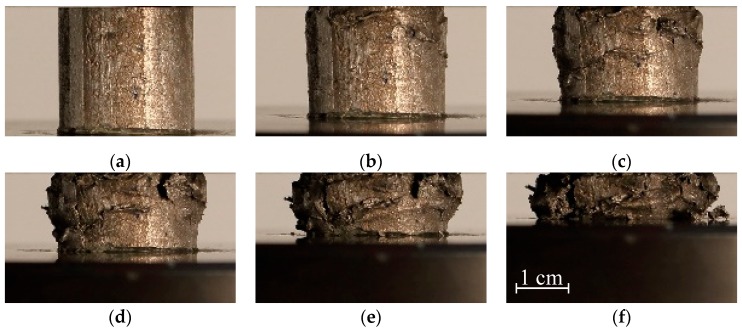
Deformation sequence of closed-cell aluminium foam under quasi-static loading conditions. Strain: (**a**) 0%; (**b**) 12%; (**c**) 24%; (**d**) 36%; (**e**) 48%; (**f**) 60%.

**Figure 5 materials-12-04108-f005:**
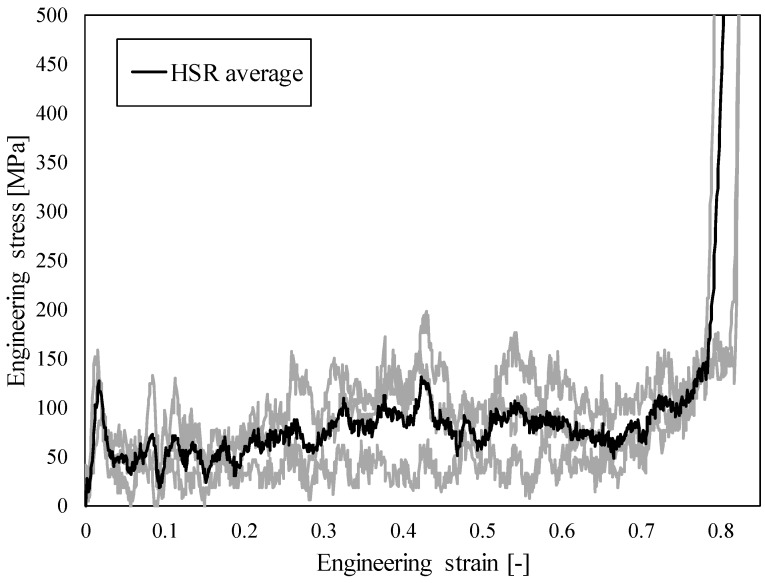
Engineering stress-strain relationship of closed-cell aluminium foam under high strain rate loading conditions.

**Figure 6 materials-12-04108-f006:**
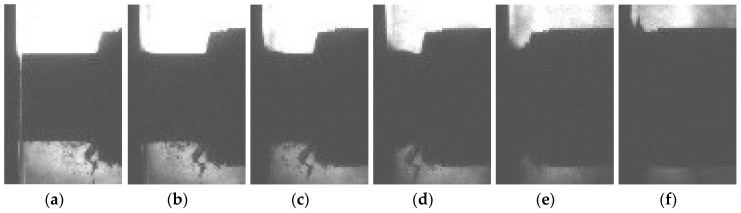
Deformation sequence of closed-cell aluminium foam under high strain rate. Strain: (**a**) 0%; (**b**) 18%; (**c**) 36%; (**d**) 54%; (**e**) 72%; (**f**) 90%.

**Figure 7 materials-12-04108-f007:**
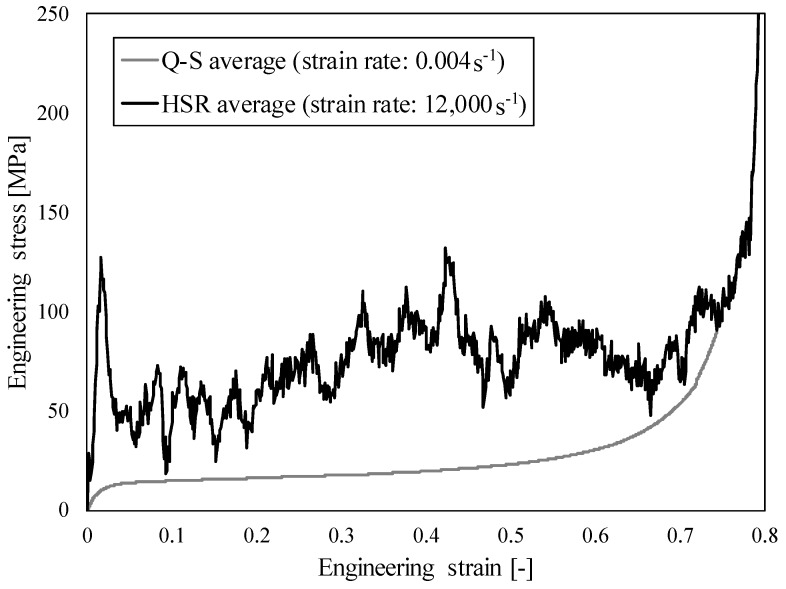
Comparison of average quasi-static and high strain rate results of closed-cell aluminium foam.

**Figure 8 materials-12-04108-f008:**
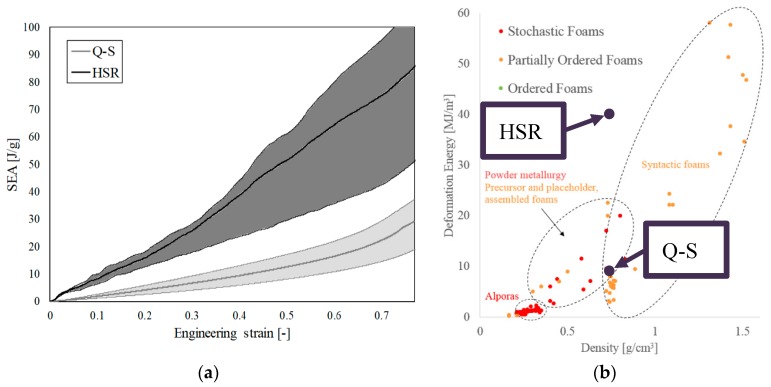
(**a**) The SEA evolution of closed-cell aluminium foam at quasi-static (Q-S) and high strain rate loading (HSR) and (**b**) comparison of the experimental results with different types of cellular metals [3].

**Figure 9 materials-12-04108-f009:**
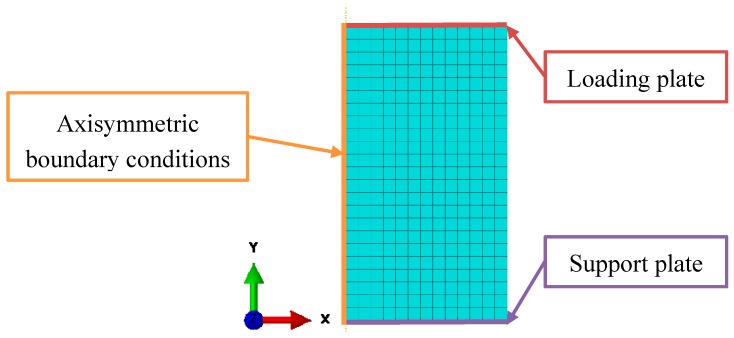
Mesh and boundary conditions of the axisymmetric computational finite element model.

**Figure 10 materials-12-04108-f010:**
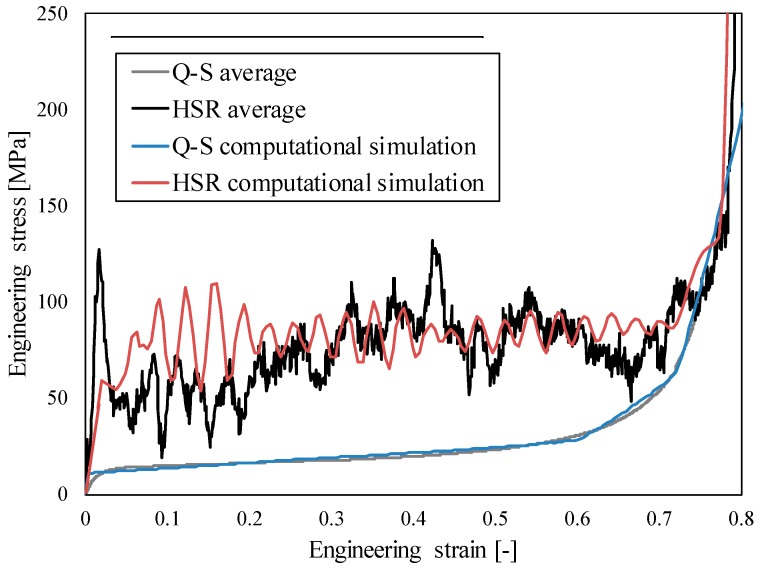
Comparison of experimental and computational results of closed-cell aluminium foam behaviour at quasi-static (Q-S) and high strain rate loading (HSR).

**Figure 11 materials-12-04108-f011:**
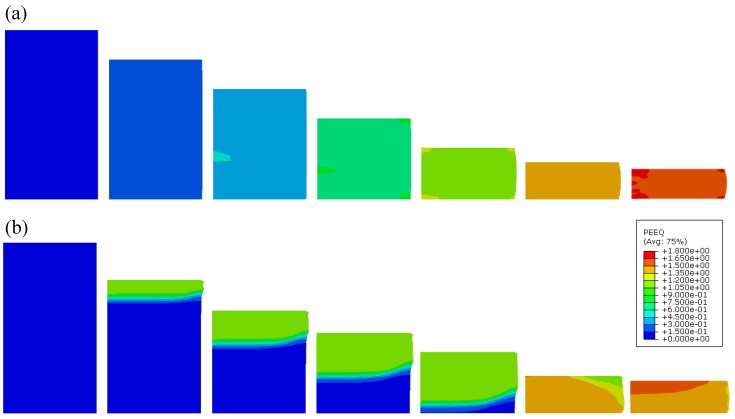
Equivalent plastic strain (PEEQ) evolution in closed-cell aluminium foam during quasi-static (**a**) and high strain rate (**b**) loading (strain increment: 15%).

**Figure 12 materials-12-04108-f012:**
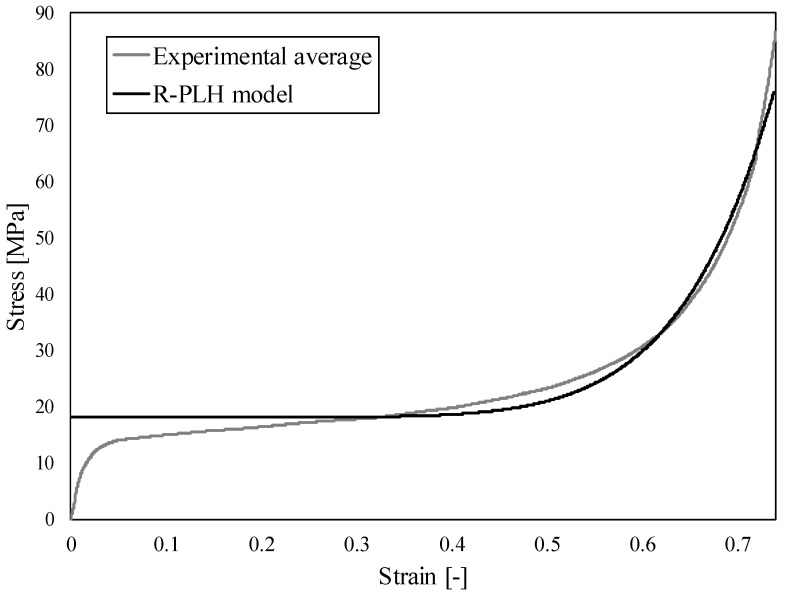
The Rigid-Power-Law Hardening (R-PLH) model fitting to average experimental results of closed-cell aluminium foam behaviour under quasi-static loading conditions.

**Figure 13 materials-12-04108-f013:**
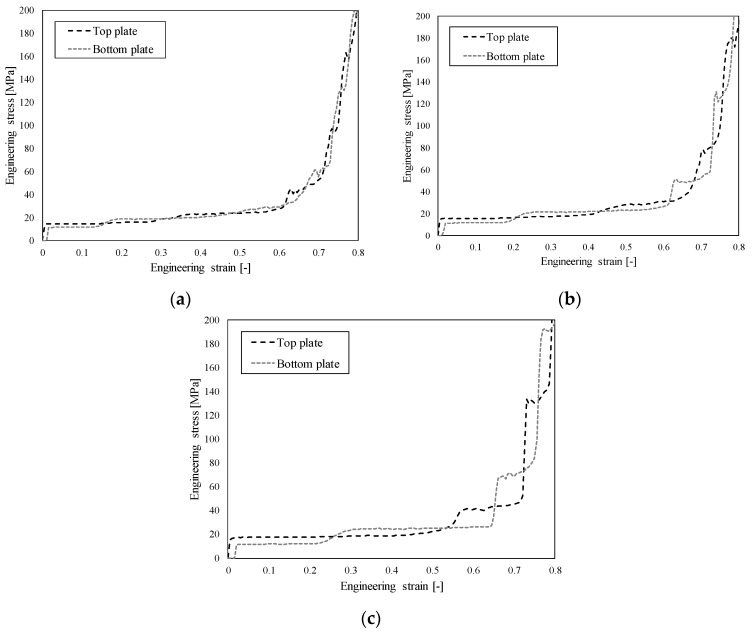
Mechanical response of closed-cell aluminium foam observed on bottom and top plate at different loading velocities. (**a**) 30 m/s, (**b**) 40 m/s and (**c**) 50 m/s.

**Figure 14 materials-12-04108-f014:**
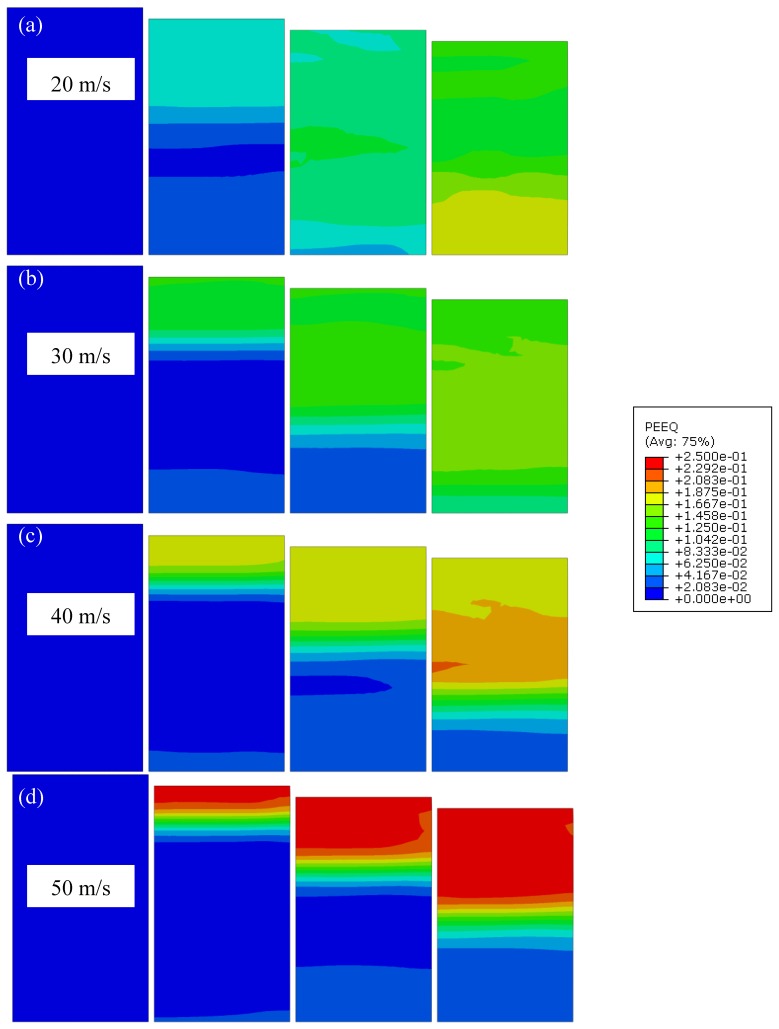
Effective plastic strain evolution (PEEQ) in closed-cell aluminium foam at different loading velocities (strain increment: 4%). Loading velocity: (**a**) 20 m/s; (**b**) 30 m/s; (**c**) 40 m/s; (**d**) 50 m/s.

**Table 1 materials-12-04108-t001:** Physical properties of the specimens (standard deviation is given in brackets).

Diameter	Length	Weight	Density	Porosity	Number of Specimens	
*d* [mm]	*l* [mm]	*m* [g]	*ρ* [kg/m^3^]	*p* [%]	Q-S Testing	HSR Testing
25.32 (0.15)	23.08 (0.23)	7.9 (0.39)	681 (0.03)	75	5	3

**Table 2 materials-12-04108-t002:** Specific energy absorption of closed-cell aluminium foam at different strain rates.

Loading Velocity Regime	SEA at 50% [J/g]	SEA at 77% [J/g]
Quasi-static	13.04	29.27
High strain rate	58.74	85.39

**Table 3 materials-12-04108-t003:** Parameters for crushable foam material model.

*ρ* [kg/m^3^]	*E* [MPa]	*ν* [–]	*k* [–]	*k*_t_ [–]
681	5023.8	0.11	1.53	0.45

**Table 4 materials-12-04108-t004:** Hardening curve definition.

Pl. Strain [–]	0	0.03	0.12	0.15	0.25	0.42	0.51	0.60	0.9	1.07	1.8
Stress [MPa]	11	11.9	15.35	16.12	18.39	20.93	23.05	25.18	29.70	69.82	2100

**Table 5 materials-12-04108-t005:** Material parameters and critical loading velocities for closed-cell aluminium foam.

Porosity *φ* [–]	*σ*_0_ [MPa]	*K* [MPa]	*n* [–]	*ε*_d_ [–]	*v_cr*1*_* [m/s]	*v_cr*2*_* [m/s]
0.75	18.1	595.7	7.7	0.5	37.5	46.1

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
