# Peer review of "Compressive Behaviour of Closed-Cell Aluminium Foam at Different Strain Rates"

_materials, 2019, doi:10.3390/ma12244108_

Round 1

Reviewer 1 Report

Interesting and well-conducted piece of research containing both experimental and theoretical parts. 

Author Response

Interesting and well-conducted piece of research containing both experimental and theoretical parts.

Reviewer 2 Report

The paper presents the experimental results, the fitting of parameters of an existing Abaqus model to evaluate the static and dynamic response of small cylinders of aluminum foam under static and hig strain rate compression.

The study is carefully done and seems sound. The authors seem to understand quite well the physics o the observed experiments. The part describing the modelling is not fully convincing. The paper could be published but still deserves improvements for the following reasons.  

The review of litterature is really insufficient and many papers in well known journels (e. g. IJSS ... or IJ I...) are not even mentionned (Kyriakides works for instance...)

In section 2 no information on the mean cell size is given. It is of importance to know how many cells in the width and height to evaluate the quality of the proposed tests. If there are to few the tests are irrealistic.

In figure 3 the strain increment is 0.12 but what is the stain for the 1rst photo?

How many dynamic tests were performed? Only 3? this is not sufficient for the evaluation of a mean response of the material...

Sections 4.n are denoted 1.n...

in section 4.1 there is a miss understanding of assoicted flow rule conditions. It is not because the yield stress depends of hydrostatic pressure that the flow rule is not associated. For instancde there exist different Drucker Prager models with or without assiociated flow rule. Please be more carefull on this point.

On which response are identified the stress strain basic curve. One may think that it is on the quasi static case..Please tell it !

How many elements in x and y directions finally? please tell it...

On which computation is the check of mesh convergence performed? It seems it has been on the quasi static one.

Section 4.2

the results seem good. But many points are debatable in the high strain rate computations:

1) the first stress peak is missed in the high strain rate. This is annoying. Please explain

2) the comparison in terms of energy absorbsion should be given

3) the model should be compared with different meshes because there is strain localisations due to wave reflections.

4) it is not because the model performs reasonably with one strain rate that it will perform well with an other one. The paper would be more covincing if at least 3 strain rates were presented.

Author Response

The paper presents the experimental results, the fitting of parameters of an existing Abaqus model to evaluate the static and dynamic response of small cylinders of aluminum foam under static and hig strain rate compression. The study is carefully done and seems sound. The authors seem to understand quite well the physics o the observed experiments. The part describing the modelling is not fully convincing. The paper could be published but still deserves improvements for the following reasons.   The review of litterature is really insufficient and many papers in well known journels (e. g. IJSS ... or IJ I...) are not even mentionned (Kyriakides works for instance...)

Additional papers according to the reviewer’s recommendation were added to the manuscript (marked with yellow):

˝Besides closed-cell foams, similar work was done in the field of open-cell foams in terms of multiaxial loading [32] and impact response [33,34].˝ In section 2 no information on the mean cell size is given. It is of importance to know how many cells in the width and height to evaluate the quality of the proposed tests. If there are to few the tests are irrealistic.

The average pore diameter of the resulting foams has been included in section 2. From the average cell size it can be concluded that more than 10 cells in each direction, confirming that the size of the specimens was representative.

In figure 3 the strain increment is 0.12 but what is the stain for the 1rst photo?

The first image illustrates the state before the loading of the specimen.

How many dynamic tests were performed? Only 3? this is not sufficient for the evaluation of a mean response of the material...

Due to high costs and complexity of the HSR experiments, 3 specimens were tested.

Sections 4.n are denoted 1.n...

The typo mistake was done during the automatic processing of the article – it has been corrected in the revised manuscript.

in section 4.1 there is a miss understanding of assoicted flow rule conditions. It is not because the yield stress depends of hydrostatic pressure that the flow rule is not associated. For instancde there exist different Drucker Prager models with or without assiociated flow rule. Please be more carefull on this point.

The sentence describing the associated flow rule in term of yield stress dependency of hydrostatic pressure was changed in the manuscript (marked with yellow).

On which response are identified the stress strain basic curve. One may think that it is on the quasi static case..Please tell it !

            Basic stress-strain curve was obtained on quasi-static case (Section 4.2):

˝The crushable foam constitutive model parameters were determined by using an optimisation algorithm, initially comparing the quasi-static experiment and computational responses…˝

How many elements in x and y directions finally? please tell it...

Global finite element size was 1 mm, which results in 13 elements in radial direction and 23 in longitudinal direction. The image of the finite element mesh was also added to Fig. 8.

On which computation is the check of mesh convergence performed? It seems it has been on the quasi static one.

As the basic stress-strain curve, also convergence study (global size of 3, 1.5, 1 and 0.75 mm) was performed under quasi-static loading conditions. Additionally, mesh sensitivity study on two meshes (global size of 1 and 1.5 mm) was done at loading velocities of 20, 50 and 270 m/s, where responses were compared visually (PEEQ) and quantitatively - reaction forces on the top and bottom plate. While only minor visual differences occurred due to ˝averaging˝ at the coarser mesh, the reaction forces were identical in both analysed cases.

Section 4.2

the results seem good. But many points are debatable in the high strain rate computations:

1) the first stress peak is missed in the high strain rate. This is annoying. Please explain

The first peak is a typical consequence of an object collision and shows an expected response of structures during an impact. Comment about that was added to manuscript (marked with yellow):

˝This stress peak is a consequence of collision in the experiments and represents a typical response of structures during the initial phase of the impact [17].˝

2) the comparison in terms of energy absorbsion should be given

The comparison of SEA in experiment and computational simulation for both loading velocities was added to the text (marked with yellow):

"Additionally, the SEA capabilities from experimental results and computational simulations were compared. The SEA values from experimental tests are given in Table 2, while the SEA values from computational simulations are 32.19 J/g in the case of Q-S loading and 95.11 J/g in the case of HSR loading. This results in 10 % over estimation of energy absorption capabilities calculated by computational simulations, which is caused mainly by discrepancy in densification region in the case of Q-S loading and due to the oscillations (no filer has been used) at plateau stress region in the case of HSR loading."

3) the model should be compared with different meshes because there is strain localisations due to wave reflections.

Response of models with different meshes were also compared at different strain rates. Please see the comment above.

4) it is not because the model performs reasonably with one strain rate that it will perform well with an other one. The paper would be more covincing if at least 3 strain rates were presented.

The authors agree that it would be meaningful to experimentally test the specimens at several strain rates. Due to cost and extensiveness of the experiments only 2 strain rates were tested. The computational models were validated at these two strain rates. Additional 4 different strain rates were analysed with computational models (Figs. 12 and 13) in order to study the strain rate hardening of closed-cell foam.

Reviewer 3 Report

The authors in this paper studied the compressive behaviour of closed-cell aluminium foam at different strain rates. The foams were produced by using the powder compact melting method. An experimental and computational approach was used. The results are interesting.

1 - In the bibliography, the authors give many information on the metal foam manufacturing approaches. Unfortunately, many of them and in particular the powder metallurgy with foaming agent are technologies that have problems on the repeatability. The authors should insert some information on this aspect. The reviewer suggests some example on authors that have studied this phenomenon:

- Barletta, M, Guarino, S, Montanari, R, Tagliaferri, V, Metal foams for structural applications: Design and manufacturing, International Journal of Computer Integrated Manufacturing, Volume 20, Issue 5, July 2007, Pages 497-504, https://doi.org/10.1080/09511920601160197 - Barletta, M, Gisario, A, Guarino, S, Rubino, G. ,Production of open cell aluminum foams by using the dissolution and sintering process (DSP), Journal of Manufacturing Science and Engineering, Transactions of the ASME, Volume 131, Issue 4, August 2009, Pages 0410091-04100910 https://doi.org/10.1115/1.3159044

2 - Authors should provide a more accurate description of the manufacturing approach used and of the experimental equipment also with a figure. Also a table summarizing all the information could be helpful.

3 - How many tests the authors have done to obtain the results? Did the experimentation provide for replications? A table summarizing all the information of experimental conditions used could be helpful

4 - Figure 2 and 5 the authors should insert the reference scale

Author Response

The authors in this paper studied the compressive behaviour of closed-cell aluminium foam at different strain rates. The foams were produced by using the powder compact melting method. An experimental and computational approach was used. The results are interesting.

1 - In the bibliography, the authors give many information on the metal foam manufacturing approaches. Unfortunately, many of them and in particular the powder metallurgy with foaming agent are technologies that have problems on the repeatability. The authors should insert some information on this aspect.

The reviewer suggests some example on authors that have studied this phenomenon:

- Barletta, M, Guarino, S, Montanari, R, Tagliaferri, V, Metal foams for structural applications: Design and manufacturing, International Journal of Computer Integrated Manufacturing, Volume 20, Issue 5, July 2007, Pages 497-504, https://doi.org/10.1080/09511920601160197

- Barletta, M, Gisario, A, Guarino, S, Rubino, G. ,Production of open cell aluminum foams by using the dissolution and sintering process (DSP), Journal of Manufacturing Science and Engineering, Transactions of the ASME, Volume 131, Issue 4, August 2009, Pages 0410091-04100910 https://doi.org/10.1115/1.3159044

- Degischer, H.P. & Kriszt, B. (Ed(s).) (2002). Handbook of Cellular Metals, Wiley-VCH, ISBN 3-527-30339-1, Weinheim.

The introduction has been revised according to the reviewer´s suggestion. Three references have been cited and included in the list of references. The references have also been renumbered.

2 - Authors should provide a more accurate description of the manufacturing approach used and of the experimental equipment also with a figure. Also a table summarizing all the information could be helpful.

The section has been revised to include an additional description and further details of the manufacturing process. Furthermore, physical properties of the specimens were given in Table 1. A new figure (Fig. 1) has also been included showing the foaming furnace, the precursor material used in this work, as well as the visual aspect of the resulting cylindrical aluminium foam and the closed-cell foam specimens for mechanical testing. The Figures have also been renumbered in the entire paper.

3 - How many tests the authors have done to obtain the results? Did the experimentation provide for replications? A table summarizing all the information of experimental conditions used could be helpful

            A table (Table 1) with properties and number of the specimens was added to the manuscript.

4 - Figure 2 and 5 the authors should insert the reference scale

            Reference scales were added to the figures in manuscript.

The authors hope to have answered satisfactorily to all reviewer's comments and suggestions. We will be happy to provide any further clarification that might be required.

Round 2

Reviewer 2 Report

the paper has been well revised and may be published though

1) the number of dynamic experiments is too low to give a reasonable basis for the identification of the model

2) the theoretical part is very classical: basic application of ABAQUS standard foam model...

Reviewer 3 Report

The paper can be accepted in the present form